Biblio-MetReS for user-friendly mining of genes and biological processes in scientific documents

Usie Anabel 1 2
Karathia Hiren 1
Teixidó Ivan 2
Alves Rui 1 ralves@cmb.udl.cat
Solsona Francesc 2 francesc@diei.udl.cat
1 Department of Basic Medical Sciences, Edifici Recerca Biomedica I, Universitat de Lleida and IRBLleida , Lleida , Spain
2 Department of Computer Science, Escola Politècnica Superior and INSPIRES, Universitat de Lleida , Lleida , Spain
Thompson Julie
Electronic publication date: 2014 Feb 27
Publication date: 2014
Volume: 2
Electronic Location ID: e276
Received 2013 Dec 4; Accepted 2014 Jan 27
Copyright: © 2014 Usie et al.
Copyright year: 2014
Copyright holder: Usie et al.
License: This is an open access article distributed under the terms of the Creative Commons Attribution License, which permits unrestricted use, distribution, and reproduction in any medium, provided the original author and source are credited.
License URL: https://creativecommons.org/licenses/by/3.0/

Keywords: Network reconstruction, Systems biology, Literature analysis

Funding: RA was partially supported by the Ministerio de Ciencia e Innovación (MICINN, Spain through grant BFU2010-17704). FS was partially funded by the MICINN, with grants TIN2011-28689-C02-02. The authors are members of the research groups 2009SGR809 and 2009SGR145, funded by the “Generalitat de Catalunya”. AU is funded by a Generalitat de Catalunya (AGAUR) PhD fellowship. The funders had no role in study design, data collection and analysis, decision to publish, or preparation of the manuscript.

==============================
One way to initiate the reconstruction of molecular circuits is by using automated text-mining techniques. Developing more efficient methods for such reconstruction is a topic of active research, and those methods are typically included by bioinformaticians in pipelines used to mine and curate large literature datasets. Nevertheless, experimental biologists have a limited number of available user-friendly tools that use text-mining for network reconstruction and require no programming skills to use. One of these tools is Biblio-MetReS. Originally, this tool permitted an on-the-fly analysis of documents contained in a number of web-based literature databases to identify co-occurrence of proteins/genes. This approach ensured results that were always up-to-date with the latest live version of the databases. However, this ‘up-to-dateness’ came at the cost of large execution times. Here we report an evolution of the application Biblio-MetReS that permits constructing co-occurrence networks for genes, GO processes, Pathways, or any combination of the three types of entities and graphically represent those entities. We show that the performance of Biblio-MetReS in identifying gene co-occurrence is as least as good as that of other comparable applications (STRING and iHOP). In addition, we also show that the identification of GO processes is on par to that reported in the latest BioCreAtIvE challenge. Finally, we also report the implementation of a new strategy that combines on-the-fly analysis of new documents with preprocessed information from documents that were encountered in previous analyses. This combination simultaneously decreases program run time and maintains ‘up-to-dateness’ of the results. Availability: http://metres.udl.cat/index.php/downloads, Contact: metres.cmb@gmail.com.

Introduction

The reconstruction of molecular circuits is an important research goal in the biological sciences. One of the ways to achieve that reconstruction starts with the use of automated text-mining techniques to identify networks of genes that co-occur in scientific documents. Subsequent human curation of these co-occurrence networks can then lead to accurate circuit reconstruction.

In this process, the automated identification of co-occurrence gene networks is crucial because databases of relevant scientific documents contain many more entries than those that can be manually analyzed (Alves & Sorribas, 2007; Markowetz & Spang, 2007; Arighi et al., 2013; Krallinger et al., 2013). A gold standard of these databases, MEDLINE, contains more than 19 × 106 records, with 2000–4000 new entries being added each day (NCBI, 2013). Extracting biological information from such large databases requires text-mining methods and tools that are able to automatically integrate and summarize useful biological information across the database records.

The development of text-mining methods that enable circuit reconstruction from scientific documents is an area of active development (Camon et al., 2005; Hoffmann & Valencia, 2005; Huang et al., 2008; Chen, Liu & Manderick, 2010; Arighi et al., 2011; Kano et al., 2011; Kim et al., 2011; Szklarczyk et al., 2011; Usié et al., 2011; Bossy et al., 2012; Kim et al., 2012; Pyysalo et al., 2012; Arighi et al., 2013; Krallinger et al., 2013). The performance of those methods for automated identification of the circuits (Camon et al., 2005), of their components (genes/proteins), and of the inter-component relationships, has been systematically evaluated over the last few years, for example through the BioNLP (Kano et al., 2011; Kim et al., 2011; Bossy et al., 2012; Kim et al., 2012; Pyysalo et al., 2012) and BioCreAtIvE initiatives (Huang et al., 2008; Chen, Liu & Manderick, 2010; Arighi et al., 2011; Wu et al., 2012; Arighi et al., 2013; Krallinger et al., 2013).

To briefly summarize, there are three large classes of methods that have been employed for the reconstruction of co-occurrence gene networks: dictionary-based methods, morphology-based methods, and context-based methods (Vazquez et al., 2011). Dictionary-based methods rely on matching compiled lists of terms to their appearances in the text of documents (Yang, Lin & Li, 2008). Morphology-based methods rely on the morphological structure of specific classes of words to single them out in documents (Malouf, 2002; Peng & Schuurmans, 2003). Finally, context-based methods can be divided into Machine Learning or Natural Language Processing techniques: The former identify patterns in the structure of the text that help to recognize the presence of the relevant entities in documents; the latter draw on our knowledge about the grammar and syntax rules of natural languages to recognize those entities. These three general approaches can be combined in order to improve NER (for example see Arighi et al., 2013; Krallinger et al., 2013 and references therein).

In general, methods participating in evaluations such as BioCreAtIvE or BioNLP are implemented in tools that can be included in web-services and assist curators in the maintenance of large databases of biological knowledge. Examples of this are given in Arighi et al. (2013), Krallinger et al. (2013). In most cases, using these methods and tools requires that one becomes an expert in computer programming.

Experimental scientists that are interested in being users of, without becoming experts in, text-mining methods to directly reconstruct networks of gene co-occurrence for their genes of interest in scientific documents have a much smaller set of available user-friendly tools. The first that became available was iHOP (Hoffmann & Valencia, 2005), which was later joined by STRING (Franceschini et al., 2013). These user friendly and intuitive web applications allow anyone to reconstruct the network of co-occurrences contained in Medline abstracts and/or Pubmed documents. Users of these applications face two important limitations. First, the applications rely on preprocessed versions of the Medline/Pubmed databases, which means that searches are fast but results are always out of date. Second, the coverage of full text documents by the applications is, at best, limited.

Recognizing these limitations, Biblio-MetReS (Bibliometric Metabolic network Reconstruction Server Usié et al., 2011) was implemented for the same target audience as STRING or iHOP, but relying on two differential features with respect to those applications. The first was that it would search databases and analyze documents on the run, thus providing the users with the most up-to-date results available on the web. The second was that full text documents were also analyzed, as were other databases besides Medline/Pubmed. These two features made Biblio-MetReS significantly slower than STRING and iHOP.

Here, we report an evolution of the application Biblio-MetReS that permits constructing co-occurrence networks for genes, GO processes, Pathways, or any combination of the three types of entities and graphically represent those entities. No other user-friendly application that we are aware of simultaneously allows the type of mixed analysis and graphical representation afforded by Biblio-MetReS. We show in a comparative analysis that the performance of Biblio-MetReS in identifying gene co-occurrence is as least as good as that of other comparable applications (STRING and iHOP). In addition, we also show that the identification of GO processes is on par to that reported in the latest BioCreAtIvE challenge (Arighi et al., 2013). Finally, we also report the implementation of a new strategy that combines on-the-fly analysis of new documents with preprocessed information from documents that were encountered in previous analyses. This combination simultaneously decreases program run time and maintains ‘up-to-dateness’ of the results.

Methods

Organism selection

Biblio-MetReS is organism-centric. Users must select their organism of interest from a list of more than 1200 organisms with fully sequenced and annotated genomes before starting any search. They must also decide whether they want to perform GO and/or pathway term co-occurrence analysis. After these decisions are made, the program loads the genes for the organism from the program’s central database. If selected, terms from the GO classification, KEGG and/or Panther pathways are also loaded into the application’s front end.

Document analysis

To analyze networks of co-occurrence Biblio-MetReS needs users to select at least one gene from their organism of interest and at least one database in which to perform document analysis. The seed list of genes is then used to identify relevant documents in the selected database(s) of documents. The text in the flagged documents is then analyzed to identify additional genes from the organism of interest, as well as GO and/or Pathway terms. This allows users to identify co-occurrence among GO/Pathway entities and between GO/Pathway entities and gene/protein entities in sentences, paragraphs or documents. For further description of this procedure, please see section 1 of the Supplementary Materials.

Biblio-MetReS uses exact matching of gene names to an internal dictionary of synonyms to identify co-mentions of genes/proteins in the text of scientific records. The gene synonyms are those officially defined by NCBI. For any given gene, all synonyms are searched for in the text of flagged documents. Similarly, Biblio-MetReS uses exact matching of GO terms to an internal dictionary of synonyms defined by the gene ontology consortium (Gene Ontology, 2013) to identify co-mentions of GO terms in the text of scientific records. The same exact matching is done to identify mentions to entities from the complete joint sets of KEGG (Kotera et al., 2012) and Panther pathways (Mi et al., 2005).

Co-occurrence of any two terms is analyzed at three levels. First, all possible pairs of terms are searched for in each document as a whole. Then, each document is divided into paragraphs and the pairs of terms identified in the document are searched for within each paragraph. Finally, each paragraph is divided into single sentences and the pairs of terms identified in that paragraph are searched for within each sentence. Within each level, the distance between each term in the pair is not taken into account.

The database containing the organisms and their gene names, as well as the GO terms, is updated every three months using information compiled automatically from NCBI and GO.

Calculating the significance of term co-occurrences

To attribute statistical significance to the co-occurrence of a pair of genes, pathway terms, GO terms, gene-pathway terms, gene-GO terms or GO-pathway terms, we calculate several metrics. First, we measure how frequently the different pairs co-occur in sentences, paragraphs and/or documents. We then take the odds ratio of the frequency of occurrences in the first two categories with respect to that of the third. The closer to one these odds ratios are, the more frequent it is that both genes are mentioned only in the same sentences or paragraphs of a document, rather than appearing haphazardly in different sections of the text.

Second, we calculate how much information we gain by having two terms, Ti and Tj, co-occur, when compared to the individual occurrences of the terms. To estimate this we use information theory. The individual probability of occurrence of a term is denoted as p(Ti) and it is formally defined as p(Ti) = a∕n, where a is the number of documents where Term Ti appears, and n is the total number of documents. The joint probability of co-occurrence of two terms, p(Ti, Tj), is defined as p(Ti, Tj) = b∕n, where b is the number of documents where terms Ti and Tj simultaneously appear, and n is the total number of documents. The mutual information, MI(Ti, Tj), is then calculated as follows: MI(Ti,Tj)=p(Ti,Tj)×log(p(Ti,Tj)∕(p(Ti)p(Tj)))

Finally, and in order to attribute some form of statistical significance to the co-occurrence of a pair of terms, we do as follows. Consider a set of n sentences (paragraphs, documents) [1, …, n]. For a given term k define yi,k=1⇐=termkoccurs∈sentence (paragraph, document)i0⇐=otherwise

Now, for terms k1 and k2 define φk1, k2 = yi, k1×yi, k2 which has value 1 when both terms co-occur and 0 otherwise.

Both these variables have a Bernoulli distribution. If the occurrence of terms k1 and k2 is independent, then p(φk1, k2) = p(yk1)p(yk2) would be expected, where p(yk, ⋅) is the relative frequency of occurrence of term yk, ⋅ and p(φk1, k2) is the relative frequency of co-occurrence of terms k1 and k2 in the total number n of sentences (paragraphs, documents). Then, a Pearson statistic can be used to test for independence of occurrence between k1 and k2 by comparing the observed frequencies, n1 = n × p(φk1, k2) and n2 = (1 − p(φk1, k2)) × n, with the expected frequencies under the null hypothesis of independence, which would be m1 = n × p(yk1) × p(yk2) and m2 = n × (1 − p(yk1) × p(yk2)). The Pearson statistic is computed as follows X2= ∑i=12(ni−mi)2mi

This statistics follows a chi-square distribution with one degree of freedom, i.e. χ12∼X2; hence, the p-value can be calculated as p=Prχ12>χ2 to assess whether the observed co-occurrence is higher than the one expected by pure chance.

Precompilation strategy

Biblio-MetReS v2.0 implements a precompilation approach that works in the following way. Any search done will identify a given number of documents in the database(s) selected by the user(s), for an organism of interest. If a given document has not been found in any previous search by any user, in the context of that organism, Biblio-MetReS will analyze it as described in section 1 of Supplementary Materials and all information contained in that document and relevant for the analysis will be stored in a central database (see section 3 of Supplementary Materials for detailed information). If a given document has been previously found by any user, its information will be directly accessed from our central database, and the document will not be reanalyzed. This means that newly found documents are mined on the fly by the program to find and count mentions of relevant entities, while mentions in documents that have been previously found are simply looked up in our central database.

Results

Biblio-MetReS and Biblio-MetReS player

Biblio-MetReS v2.0 can be used to identify genes/proteins from more than 1200 different organisms in records stored in a variety of databases. Users download the application and run it locally. A functioning internet connection and a local copy of JAVA are required for the program to work. Upon starting the program, users log into the central Biblio-MetReS database and choose which organism they are interested in and whether they want to search only for co-occurrence of genes/proteins or if they also want to include biological pathways and/or GO biological processes in the analysis. Once this choice is made, the program loads the necessary information from the central database. Taking this approach, instead of including all the data locally in the program installation, permits making an application that is much smaller in size and needs less RAM to function properly. Subsequently, users select the source of documents that they want to analyze, as well as the genes/proteins and/or pathways/GO biological processes that they want to search for. They can also include their own handpicked list of processes to be searched. Once the search is launched, Biblio-MetReS will identify documents in the relevant databases that contain mentions to the relevant search items. After identifying these documents, the application fully analyzes them to identify mentions for any additional gene or Pathway/GO biological process via a dictionary matching approach. The co-occurrence of the different entities is analyzed at the level of the whole document, of individual paragraphs and of individual sentences, and the significance of this co-occurrence is calculated as described in Usié et al. (2011). The information that is relevant for the co-occurrence calculations is stored in the central Biblio-MetReS database. Any subsequent searches that identify the same document will not reanalyze it; instead, these numbers are directly retrieved from that database. Once the analysis is complete, the users can visualize it in graphical and textual form. Links to the documents and sentences where co-occurrences are found are provided. Graphical visualization of the results can be done in different ways. Users can create graphs for the global co-occurrences network and for gene- or pathway/process-centric co-occurrences at the document/paragraph and sentence levels. Significance and Mutual Information of each co-occurrence is also provided in tabular form. The graphical representation of the networks is automatically stored in local xml files. These files can be opened using a small app, Biblio-MetReS Player, which can be downloaded from the Biblio-MetReS website. This permits reviewing previously obtained networks without having to redo the search. All this process is summarized in Fig. 1.

Figure 1 User workflow for Biblio-MetReS.

Comparative performance and benchmarking of new types of entities

The benchmarking of the program and its improvements with respect to version 1.0 was carried out using four organisms of interest: Saccharomyces cerevisiae, Homo sapiens, Escherichia coli and Drosophila melanogaster. For each organism we used as a search seed a set of genes belonging to Glycolysis, Lysine metabolism and RNA processing pathways. The genes were chosen to reproduce the experiments reported in Usié et al. (2011). Details are shown in Table S1. We benchmarked three different aspects of Biblio-MetReS. First, we benchmarked the comparative identification of genes between Biblio-MetReS, iHOP, and STRING, given that these three applications have similar target audiences. Second, we benchmarked the ability of Biblio-MetReS to identify biological processes/pathways. Finally, we benchmarked the improvements in Biblio-MetReS run time made by implementing the combined pre-processing/live analysis strategy.

Benchmarking the comparative identification of genes between Biblio-MetReS, iHOP, and STRING was done in the following way. We used the genes and organisms described in Table S1 to interrogate independently Biblio-MetReS, iHOP, and STRING. For each of the three applications, the complete set of results for each gene from the same pathway were pooled together for analysis. The results from STRING were further filtered to eliminate all genes and interactions that were not literature based. Table S2 compares the results for the three applications. In summary, Biblio-MetReS find the largest number of genes, followed by STRING, and iHOP. The number of genes found by STRING and Biblio-MetReS are of the same order of magnitude, while iHOP finds between one and two orders of magnitude less genes. This result derives from the fact that iHOP analyzes only Medline abstracts, while Biblio-MetReS and STRING analyze the full text of Pubmed publications, in addition to the Medline abstracts. This is confirmed by the fact that, when Biblio-MetReS is run only to analyze Medline abstracts, it finds a similar number of genes as iHOP (data not shown). As was observed in Table S3, the genes found by each of the three applications for the same experiments only partially overlap and this is explained by the different datasets analyzed by each of the programs and by partially different dictionaries of gene synonyms (Usié et al., 2011).

Neither STRING nor iHOP permit identifying GO terms and their associations to genes. Therefore we cannot perform experiments that are similar to the comparative benchmarking experiments described above. In light of this, benchmarking of Biblio-MetReS’ ability to identify biological processes/pathways was done in the following way. To perform the GO identification benchmark we used the test and development sets of the BioCreAtIvE IV GO task corpus [3]. We used Biblio-MetReS dictionary matching approach to identify GO terms in the non-annotated documents and then analyzed the corresponding annotated documents. We found that Biblio-MetReS identified 100% of the annotated GO terms in both sets (2963 terms in the training set and 2243 terms in the development set). Biblio-MetReS also identifies 2259 additional GO terms in the development set and 2119 additional GO terms in the training set. For the purpose of our testing these terms must be considered false positive. Taking this into account, the precision in the development set is 50%, while in the training set it increases to 58%. The F-score performance of Biblio-MetReS is 33% in the development set and 37% in the training set, which is on par with the best approaches presented in the lattest BioCreAtIvE IV challenge (Mao et al., 2013).

Benchmarking run time was done in two ways. First, we search only the Pubmed database. Second, we search by selecting all literature databases available in Biblio-MetReS. In both tests we used all the seed genes from Table S1. Each seed is used by Biblio-MetReS as a query search. This query search is launched twice. When the first search is done there are no preprocessed documents in Biblio-MetReS’ database. The information in documents is analyzed on-the-fly and stored. Then the searches are repeated, now with the documents stored in Biblio-MetReS’ database. This allows us to estimate the percentage of run-time saved by preprocessing the documents. The results are shown in Fig. 2 and in Table S2 of the supplementary materials. On average we get decreases in run time of more than 90%.

Figure 2 Effect of preprocessing documents on Biblio-MetReS’ run time.

In brief, genes from three KEGG-defined pathways are used for this test. Panels A.x show experimental results for glycolysis genes. Panels B.x show experimental results for Lysine metabolism genes. Panels C.x show experimental results for RNA processing genes. Three organisms are used in this benchmark. Panels Y.1 show results for Homo sapiens, panels Y.2 show results for Drosophila melanogaster, panels Y.3 show results for Escherichia coli, and panels Y.4 show results for Saccharomyces cerevisiae. These pathways and organisms were chosen to remain consistent with the tests performed in Usié et al. (2011). Searches were done selecting all the databases in the application. Graphs can be interpreted as follows. Light gray bars indicate the run time for Biblio-MetReS when the corresponding gene is searched for the first time. In this case the program has to do a full document analysis on the fly and no information has been preprocessed. Darker gray bars indicate the run time for Biblio-MetReS when the search for the corresponding gene is repeated, and preprocessed information is already present in Biblio-MetReS’ central database. The column ‘All’ indicates the run-time for searching all genes in the graph simultaneously, after individual searches for each gene had already been done and results preprocessed and stored.

Discussion

Here we present the new version of Biblio-MetReS, a user friendly tool for the identification of gene/protein co-occurrence networks in scientific documents. The major changes with respect to version 1.0 have to do with the search and analysis process of the documents, which can now be up to 95% faster than in the previous version. In addition, the tool now also searches for co-occurrences of biological processes and pathways, to help users to more easily establish the biological circuits in which their genes of interest may be involved in.

The methods used by the application to identify genes and proteins, as well as biological processes and pathways, in the documents are dictionary-based. These methods perform on par with iHOP and STRING for gene and protein identification and with the best BioCreAtIvE methods for biological process identification (see Supplementary Materials).

Taken together, the new application further facilitates the identification of functional relationships between proteins and aids in identifying the biological processes and circuits in which those proteins may be involved. Although GO term search has been implemented in several literature search tools (see for example Doms & Schroeder, 2005; Plake et al., 2009, among others), no other user-friendly tool permits simultaneous graphical reconstruction of networks of co-occurrence between genes, GO terms, and Pathway terms.

As is demonstrated by the BioCreAtIvE challenge (Mao et al., 2013), the problem of identifying entities in scientific texts is far from solved. Although Biblio-MetReS aims at giving non-expert users the possibility of performing such identification and use that identification to extract biological knowledge, there is much room for improvement. We are implementing an offline system to automatically search, analyze, and store information about gene/protein and pathway/biological processes co-occurrences in the documents. This will contribute to decrease the dependence of Biblio-MetReS on the users and their searches to preprocess information and make searches faster.

Supplemental Information

Supplemental Information 1 Click here for additional data file.

Additional Information and Declarations

Competing Interests

Author Contributions

The authors declare that they have no competing interests.

Anabel Usie and Rui Alves conceived and designed the experiments, performed the experiments, analyzed the data, contributed reagents/materials/analysis tools, wrote the paper.

Hiren Karathia and Ivan Teixidó contributed reagents/materials/analysis tools.

Francesc Solsona conceived and designed the experiments, analyzed the data, contributed reagents/materials/analysis tools, wrote the paper.

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
