# Peer review of "Biblio-MetReS for user-friendly mining of genes and biological processes in scientific documents"

_PeerJ, doi:10.7717/peerj.276_

## Round 0.1 · original submission · Major Revisions

The reviewers have both suggested that the technical improvements in this version of Biblio-MetReS could be described more clearly. The important criticisms of both reviewers should be taken into account in a revised version of the manuscript.

Reviewer 1 ·

Basic reporting

The authors describe an update of their approach 'Biblio-MetReS'. A first description of the tool was published in 2011. The update the authors made in comparison with the prior publication are
• the inclusion of additional recognition classes (GO names and KEGG and Panther pathway names).
• Combination of on the fly analysis with preprocessing of previously analysed documents

I propose to reject the publication because it is not ‘self-contained,’ .
The communication of a performance increase in processing is not enough for a publication. The authors should think of a version that can be accessed directly over the web. If such a tool could be used to search in the content and in addition could induce new searches it would be an appropriate unit of publication. The direct accessibility of the content over the web would be much more user friendly because no software has to be installed and handled.
For the inclusion of additional dictionaries I would propose to integrate some application scenarios emphasizing the impact of the new dictionaries.

Additional comments

Molecular circuits are a weird expression for co occurrence networks; since you do only co occurrence you should use other expressions
Please cite the main biocreative overview papers for gene mention and gene normalisation.

The English has to be improved; please contact a native speaker to revise the manuscript

Go process has already been integrated in other search engines!
Take a look at gopubmed and gogene:

http://projects.biotec.tu-dresden.de/gogene/

Experimental design

...

Validity of the findings

...

·

Basic reporting

A more recent set of bibliography about text-mining methods can give us an idea of the real activity on the area, and about the need for a new tool.

Methods are not described well enough in the main text neither in the supplementary material. A description of how the co-occurrence method is working. What is the minimum distance between terms in a sentence to be considered related? Who is the main term and who the synonyms? How that list is compiled? Are you using semantic similarity to explore the hierarchy of the Gene Ontology or KEEG? What sort of statistics have been applied? or How mutual-information is calculated? are questions that must be answered. If all this questions are solved in the previous work, then, It has to be better referenced.

In the structure of the text, from line 101 to line 134 are placed as Results but in my opinion should be part of the methods because authors are describing how the tool works and not results about the improvements of the tool.

In results I miss an overview of the accuracy statistics. Values of sensitivity and specificity of the method. Deviation from the random signal.

In Line148, figure 1 have be changed to figure 2.

Experimental design

The experimental design is correct.

Validity of the findings

The paper describes a technical improvement, but not a scientific innovation. Only considering technical innovation, the improvements that are described are agree with the results showed.

Additional comments

I consider that authors have to emphasize the improvement in biological processes and pathways annotations That is the real finding. Computational optimizations have to be described mainly in the web page of the tool and briefly in the article.

---

## Round 0.2 · accepted · Accept

The reviewers are happy with your modifications and consider that your article is now suitable for publication.

·

Basic reporting

Authors have covered all the previous comments and the present work is good enough to be published.

Experimental design

OK

Validity of the findings

Is valid

Additional comments

I consider that work is acceptably good to be published in this journal